# Genome-Wide Characterization and Expression Profiling of GASA Genes during Different Stages of Seed Development in Grapevine (*Vitis vinifera* L.) Predict Their Involvement in Seed Development

**DOI:** 10.3390/ijms21031088

**Published:** 2020-02-06

**Authors:** Bilal Ahmad, Jin Yao, Songlin Zhang, Xingmei Li, Xiuming Zhang, Vivek Yadav, Xiping Wang

**Affiliations:** 1State Key Laboratory of Crop Stress Biology in Arid Areas, College of Horticulture, Northwest A&F University, Yangling 712100, China; bajwa1999@nwafu.edu.cn (B.A.); jin.yao@nwafu.edu.cn (J.Y.); zhangsonglin@nwafu.edu.cn (S.Z.); 18838933960@163.com (X.L.); zhangxiuming00@126.com (X.Z.); 2Key Laboratory of Horticultural Plant Biology and Germplasm Innovation in Northwest China, Northwest A&F University, Ministry of Agriculture, Yangling 712100, China; vivekyadav@nwafu.edu.cn

**Keywords:** bioinformatics, grapevine, ovule abortion, *VvGAST*, GASR, *Cis*-elements

## Abstract

Members of the plant-specific GASA (gibberellic acid-stimulated Arabidopsis) gene family have multiple potential roles in plant growth and development, particularly in flower induction and seed development. However, limited information is available about the functions of these genes in fruit plants, particularly in grapes. We identified 14 GASA genes in grapevine (*Vitis vinifera* L.) and performed comprehensive bioinformatics and expression analyses. In the bioinformatics analysis, the locations of genes on chromosomes, physiochemical properties of proteins, protein structure, and subcellular positions were described. We evaluated GASA proteins in terms of domain structure, exon-intron distribution, motif arrangements, promoter analysis, phylogenetic, and evolutionary history. According to the results, the GASA domain is conserved in all proteins and the proteins are divided into three well-conserved subgroups. Synteny analysis proposed that segmental and tandem duplication have played a role in the expansion of the GASA gene family in grapes, and duplicated gene pairs have negative selection pressure. Most of the proteins were predicted to be in the extracellular region, chloroplasts, and the vacuole. In silico promoter analysis suggested that the GASA genes may influence different hormone signaling pathways and stress-related mechanisms. Additionally, we performed a comparison of the expression between seedless (Thompson seedless) and seeded (Red globe) cultivars in different plant parts, including the ovule during different stages of development. Furthermore, some genes were differentially expressed in different tissues, signifying their role in grapevine growth and development. Several genes (*VvGASA2* and *7*) showed different expression levels in later phases of seed development in Red globe and Thompson seedless, suggesting their involvement in seed development. Our study presents the first genome-wide identification and expression profiling of grapevine GASA genes and provides the basis for functional characterization of GASA genes in grapes. We surmise that this information may provide new potential resources for the molecular breeding of grapes.

## 1. Introduction

Snakins are plant antimicrobial peptides (AMPs) of the GASA (gibberellic acid-stimulated Arabidopsis) gene family. These peptides have varied functions in response to various biotic and abiotic stresses via hormonal crosstalk [1] Snakin/GASA/ GAST (GA-stimulated transcripts)/GASR (GA-stimulated regulator) is a cysteine-rich low molecular weight peptide and part of the gibberellin-regulated family [2]. Mostly, hormone-regulated gene families affect various physiological processes and plant development. Various developmental roles have been speculated for GASA genes such as lateral root initiation and development, leaf expansion, flower induction, fruit cell size regulation, seed development and germination in many monocot and dicot plants [3,4]. Apart from these, most of the GASA genes are involved in hormonal (gibberellic acid, abscisic acid, and naphthalene acetic acid) signaling pathways [5,6]. For example, in rice, *OsGSR1* (a member of GASA gene family) influences Brassinosteroid signaling by interacting with *DIM*/*DWF1* [7].

The GASA gene family is highly specific to plants; the name was assigned according to the first identified member GAST-1 from tomatoes [2]. GASA genes encode low molecular weight proteins (80–270 amino acids) and have three different domains: (1) a N-terminal signal peptide of 18–29 amino acids; (2) a highly variable region (7–31 amino acids) displaying a difference between family members both in amino acid composition and sequence length; and (3) a C-terminal GASA domain of 60 amino acids with 12 conserved cysteine residues that contribute to the biochemical stability of the molecule [3,8]. In Arabidopsis, AtGASA2/AtGASA23, AtGASA5, and AtGASA14 are involved in ABA signaling [9]. Some GASA members may have opposite functions, e.g., AtGASA5 inhibits flowering while AtGASA4 promotes flowering [3,10,11]. Furthermore, GASA family members also play a role in disease resistance; for example, in rubber plants, GASA genes (HbGASA) were up-regulated upon encounter with the fungal pathogen *Colletotrichum gloeosporioides*. The HbGASA gene induced the production of reactive oxygen, signifying its role in plant innate immunity [12]. In *Solanum tuberosum* subsp. tuberosum cv. Kennebec, the expression of snakin1, 2, and 3 was found to be affected by the inoculation of bacterial fungal pathogens [13]. 

In the recent past, several studies focused on the functional characterization of these low molecular weight peptide-proteins in different plant species such as *Arabidopsis*, tomato, rice, potato, maize, wheat, apple, soybean, rubber plant, gerbera, strawberry, French bean, beechnut, pepper, and petunia [5,12,13,14,15,16,17,18]. Different studies found that GASA genes have potential roles in flower induction, seed size, seed development, and fruit size regulation, e.g., in Arabidopsis, overexpression of GASA4 positively affected seed size, seed weight, and seed yield [11]. Likewise, another member of this gene family, TaGASR7, has been found to be associated with grain length in wheat [19,20]. Similarly, application of different growth regulators (i.e., GA3, 6-BA, and sugar) in apple showed that GASA genes may have a role in flower induction [17]. While working with rice, Muhammad et al. [21] found that GASA genes increased grain size and length. Little information is available about the functions of GASA genes in fruit plants; however, for grapevine GASA genes; this will be the first study to provide such information.

Grapevine (*Vitis vinifera* L.) is among the top fruits crops grown all over the world, with an annual production of 7.9 million ha globally [22]. The cultivation is important due to its multipurpose use including table grapes, juice, raisins, and wine production. Grapes are supposed as an ideal model plant to study and understand the berry development phenomenon in perennial fruit crops. Recently, the demand for seedless grapes is also increasing [23,24]. In recent years, scientists have reported genes such as *VvYABBY4* and *VvHB58* as having a role in grapevine fruit and seed development [25,26]. However, the key genes mediating this process still needs to be explored, such as, for example, GASA genes. The important role of GASA genes as a regulator of different stages of plant growth, especially in flower induction, seed size, and seed weight in other fruits, justifies a detailed bioinformatics and expression profiling of this gene family in grapes. In this experiment, we performed a detailed bioinformatics study of the GASA gene family in grapes, including chromosomal locations and gene structure, sequence homology, evolutionary history, synteny analysis, cis-acting element analysis, in silico analysis of protein structure, and subcellular localization. We also investigated the expression of GASA genes during different phases of seed development as well as in different tissues in seedless and seeded grape cultivars. The findings of this experiment will provide foundations for further detailed studies of GASA genes in grapes as well as in other fruit plants. 

## 2. Results

### 2.1. Genome-Wide Identification and Protein Features of GASA Genes in Grapevine

A total of 14 putative GASA genes were identified in the grapevine genome. These genes were named according to their locations on the chromosomes. Complete information about grapevine GASA genes is presented in Table 1. 

The protein sequence of VvGASA genes varied from 64 (*VvGASA13*) to 298 (*VvGASA5*) amino acids with a MW (molecular weight) of 7.28 to 31.96 kDa. The average length of grapevine GASA proteins was 109 aa, while the average MW was 12.06 kDa. Apart from these, the isoelectric point (PI) ranged from 8.50 (*VvGASA13*) to 9.64 (*VvGASA5*), while for most of the proteins (72%), instability index values were more than 40. According to the Grand average of hydropathicity (GRAVY), the GASA proteins are hydrophilic except for *VvGASA8* and *VvGASA14*. As far as the amino acid content of proteins was concerned, cysteine, lysine, and leucine were predominant amino residues, whereas the aliphatic index ranged from 33.43 to 81.79. Detailed information about protein characteristics can be seen in Table 2.

Prediction of the subcellular positions of proteins can give important hints about their roles. From in silico analysis, the subcellular locations and structures of the proteins were determined. Most of the grapevine GASA genes were found in the apoplast (cell wall), vacuole, chloroplast, and cytoplasm (Table 2). All proteins of GASA genes have flexible structure due to the presence of coils. Members of the group 1 GASA gene family have more coils compared to other groups as shown in Figure 1. All proteins have at least two large α helices while β sheets are not common. Apart from two large α helices, group 1 proteins and GASA12 also have two small α helices.

### 2.2. Phylogenetic Analysis of GASA Genes from Grape, Apple and Arabidopsis

To describe the phylogeny and to assist in the classification of the GASA gene family in grapevine, a phylogenetic tree was constructed among grapes, apple, and *Arabidopsis*-aligned GASA protein sequences. The analysis included 55 GASA genes comprising 14, 26, and 15 from grapes, apple, and *Arabidopsis*, respectively. As shown in Figure 2, the genes were divided into three groups named G1, G2, and G3. In the distribution of VvGASA genes into different groups, the previous trend was noted, i.e., G2 contained the least (three) VvGASA genes, while G3 contained the most (six) genes. Grapes and *Arabidopsis* have the same number of genes in G1 and G3, while G2 has a difference of one gene. The predicted length of grapevine proteins in the G1, G2, and G3 groups ranges from 64–106, 67–88, and 74–298 amino acids, respectively.

### 2.3. Analysis of Conserved Motifs, Domain Architecture, and Gene Structure 

To further explore the phylogeny of grapevine GASA genes, an unrooted tree was constructed between VvGASA genes (Figure 3A). In concordance with the phylogenetic tree including the *Arabidopsis*, grapes, and apple GASA genes, this analysis also supported the classification of GASA genes into three groups. The exon–intron structural analysis of VvGASA genes was performed by the Gene Structure Display Server program to gain some perceptible information about the paralogous genes. Each member of G2 contained two exons (Figure 3C), showing that this group is structurally more conserved as compared to the others. In G1, three members (*GASA4*, *6*, and *7*) have four exons, while the remaining two have two exons per gene. Meanwhile, in G3, three members out of six have the same (four) exon number, and others have a different number of exons, showing that G3 is less conserved. The genes with similar exon numbers, positions, and lengths were closely related paralogous gene pairs. Moreover, the similarity index of protein sequences varied in each clade, with 59.4%, 57.4%, and 20.23% in G1, G2, and G3 groups, respectively. Furthermore, the similarity index of all grapevine GASA gene was 16.87%. This suggests that not only the exon number but also the protein similarity index is conserved with in the same clade. The motif distribution pattern was determined using an online server (MEME). GASA protein sequences have variations in motif length and number but have similar motif distribution patterns within the same group (Figure 3B). The highly conserved motifs 1 and 2 (gibberellin regulated protein (GASA domain); IPR003854) were detected in all fourteen genes of the GASA family, whereas, except for two members (GASA8 and 13), all members have motif 3. 

Moreover, the predicted number of motifs in the G1, G2, and G3 groups ranges from 2–5, 3–4, and 3–5 motifs, respectively. Therefore, these results strongly support G2 as being more conserved with respect to motif number as compared to other groups. According to the study, for members of the same clade of phylogenetic tree especially, paralogous gene pairs (*VvGASA6*/*VvGASA7* and *VvGASA*1/*VvGASA8*) shared an almost similar motif distribution either with respect to gene length or motif number. Some of the identified motifs were specific to only paralogous gene pairs; for example, motif 5 presented only in *GASA6* and *GASA7* while motif 6 was present only in *GASA1* and *GASA8*. The protein motifs that are limited to only one *VvGASA* group may have some special functions. These results further verify our classification and justify the credibility of phylogeny and exon-intron analysis for classification. To further explore the phylogenetic relationships among grapevine GASA genes, the presence of the GASA domain was examined in all genes. For this, full length VvGASA proteins were aligned. The conserved domain in *VvGASA* sequences were confirmed with SMART and multiple sequence alignment. All of the putative VvGASA proteins shared a conserved GASA domain on the C-terminal comprising about sixty amino acids with twelve cysteine residues (Appendix A).

### 2.4. Grapevine Genes Duplication and Evolutionary Analysis

According to our results, 14 *VvGASA* genes were randomly distributed on eight out of the 20 chromosomes (Figure 4). Chromosome 18 has the larger proportion of GASA genes (4; 35%). According to the criteria mentioned in Materials and Methods, four genes are tandemly duplicated (Table 3) and clustered by two duplication events on Chromosome 18 (GASA11 and 12) and an uncharacterized chromosome (GASA13 and 14). These duplicated genes belong to groups 1 and 3, and no tandem duplication was observed in group 2. Apart from tandem duplication, four pairs (VvGASA3/9, VvGASA7/6, VvGASA8/5, and VvGASA9/2) (Figure 4, Appendix A) of segmental duplication were also observed between seven genes.

This indicates that tandem and segmental duplication both have contributed in the expansion of the GASA family in grapevine. Interestingly, *VvGASA9* paired with two genes *GASA3* and *9*. All pairs of duplicated genes (segmental or tandem) “belonged to the same group suggesting common ancestor”. In conclusion, 78.5% of GASA genes (11 out of 14) underwent duplication events, which may provide clues for the expansion and functional potential of the GASA gene family. The ratio between the non-synonymous (Ka) and synonymous (Ks) can be used to describe the history of the evolutionary process [27]. The ratio between Ka and Ks was calculated for duplicated gene pairs. All duplicated gene pairs (tandem and segmental duplication) have Ka/Ks values less than 1, which suggests purifying selection, whereas the average of Ka/Ks values were 0.565 and 0.224 in tandemly and segmentally duplicated gene pairs, respectively. According to these results, segmentally duplicated genes are more conserved compared to tandemly duplicated genes.

### 2.5. GASA Genes Expression Profiling During Seed Development

To identify whether some GASA genes have a role in ovule abortion or seed development, we performed real-time, quantitative RT-PCR of GASA genes during different phases of seed development in seeded and seedless cultivars. 

As shown in Figure 5, GASA2, GASA4, and GASA11 were highly expressed during all stages of seed development in seeded cultivars compared to seedless cultivars, whereas GASA6, GASA7, and GASA8 showed expression in the later three seed developmental stages (34, 40, and 50 DAF (days after full bloom)) in the Red globe. However, GASA5 was significantly highly expressed in seedless cultivar compared to seeded grape cultivar. These results suggest that the above-mentioned differentially expressed genes may have a role in ovule abortion or seed development. 

### 2.6. Tissue Specific Expression Profiling of Grapevine GASA Candidates

The spatio-temporal expression analysis of genes can provide information about gene function [28]. We performed real-time RT-PCR for expression profiling of the grapevine GASA genes in the leaf, tendril, stem, flower, and fruit of the Thompson seedless and Red globe (Figure 6). 

We noted that some of the genes (VvGASA2 and VvGASA11) are expressed relatively ubiquitously. However, most of the GASA genes showed different levels of expression in all tissues both in seeded and seedless cultivars. For example, VvGASA7 and VvGASA8 had greater expression in all tissues (except fruit) of Red Globe, whereas in tissues of Thompson seedless, a moderate level of expression of these genes was noted, suggesting their role in seed development. In general, most of the genes were highly expressed in vegetative plant parts (leaf, stem, and tendril) compared to reproductive organs (flower and fruit), suggesting their role in plant development.

### 2.7. Promoter Analysis of GASA Genes

To further explore the regulatory mechanisms of grapevine GASA genes, in silico promoter analysis was performed (Figure 7). Several plant hormone (P box, ERE, CGTCA, ABRE, AuxRR-core, TGA-element, GARE, TCA element, and SARE)-related cis-elements were identified in the promoter region of VvGASA genes. However, there were more cis-elements related to ethylene, gibberellic acid and salicylic acid. Cis-elements related to different types of stresses (LTR, STRE, and TCA-motif) and disease resistance (W-box, WUN, WRKY, and TC-rich repeats) were identified in most of the genes. Apart from these, cis-elements involved in endosperm expression (AAGAA-motif) and meristem activation (CCGTCC-box) were identified in the promoters of 10 genes. Moreover, cis-elements having role in anaerobic respiration (ARE) and light response (BOX4, GATA, G BOX, and I BOX) were found in the promoters of all GASA genes.

## 3. Discussion

GASA gene family members have different critical roles in plant growth and development by influencing plant hormone levels and signal transduction pathways [29]. The members of the same GASA gene family may have the same or reverse functions during vegetative and reproductive growth. Various reports have mentioned negative correlations among GASA gene family members with regard to their function, e.g., overexpression of *AtGASA5* inhibited stem elongation and delay in flowering time, but overexpression of *AtGASA6* promoted early flowering [11]. Moreover, the co-suppression of *GASA4* and *GASA6* in *Arabidopsis* causes a delay in flowering time. Due to these complexities in the functional mechanisms, little information about the exact or more precise functions of GASA genes are available [3,11,30,31]. *PpyGAST* genes influenced bud dormancy by participating in GA biosynthesis and ABA signaling pathways [32]. In strawberry, synergistic action of GAST1 and GAST2 affected fruit cell size, suggesting their role in final fruit size determination [33]. However, according to our information, this study represents the first comprehensive genome-wide identification and expression profiling of GASA genes in grapevine.

In this experiment, we identified 14 GASA genes in grapes, performed comprehensive bioinformatics and expression analysis in different plant structures as well as for different stages of ovule development in seeded and seedless cultivars. The identified genes were divided into three subgroups (G1, G2, and G3) based on their phylogenetic analysis with other species including *Arabidopsis* and apple. According to phylogenetic analysis of GASA genes, grape is more phylogenetically related with *Arabidopsis*. As far as the number of genes in subgroups are concerned, we observed the previous trend [17,18]: Group 3 contained the most (six) genes, whereas G2 contained the least (three) number of genes. According to this study, group 2 is more conserved with regard to exon-intron numbers or conserved motifs, suggesting that during the evolutionary process, other groups (G1 and 3) have either gained or lost exons leading to the difference in the number of exon-introns. 

Gene duplication, which has played a major role in the evolution of gene families, can take place through four mechanisms: whole genome duplication, tandem duplication, segmental duplication, and transposition events [34]. However, segmental and tandem duplication have contributed more to the expansion and functional divergence of gene families [35]. According to our findings (Table 3), both segmental and tandem duplications contributed in the evolutionary process of grapevine GASA genes. The result of this study corroborates the previous findings that segmental duplications (7 out of 14 genes) has occurred more frequently compared to tandem duplication (4 out of 14 genes). We noticed an uneven distribution of genes on different chromosomes, as chromosome 18 contained 35% of genes. These findings suggest that duplication of GASA genes has occurred on chromosome 18 during the expansion of the grapevine GASA gene family. This finding is also supported by our observation that most of the (3 out of 4) genes on chromosome 18 underwent duplication and divided into two different groups (G2 and G3).

The Ka/Ks ratio can provide information about phylogenetic reconstruction, evolutionary process, and selection pressure. The evolution of new genes take place due to selection and mutation [36]. According to our results, the Ka/Ks values of all duplicated gene pairs were less than 1, suggesting negative (purifying) selection. Therefore, we can predict that grapevine GASA genes are less exposed to environmental changes. The analysis of the promoter region of a gene can provide clues related to its function and assist in functional characterization [37]. The presence of GA-responsive elements (P Box and GARE) in all members of group1, suggests their role in GA signaling pathways and seed development. Apart from hormone- and disease-related motifs, the presence of motifs involved in endosperm (AAGAA-motif) expression and meristem activation (CCGTCC-box) in the promoters of 10 genes suggests that GASA genes are involved in complex regulatory mechanisms affecting the expression of a gene. 

Expression profiling of genes in different plant parts and organs can provide important clues for their functional characterization. In different plant species, GASA genes have shown their spatiotemporal specificity, probably due to their involvement in different hormonal signaling pathways [7,18]. In previous findings, exogenous application of GA increased expression of GsGASA1 in leaves but down-regulated its expression in roots of the soybean [6]. However, in Arabidopsis, GA up-regulated GAST1 expression in meristem tissues but showed negative results in roots and leaves. These results showed that GASA genes have tissue specific responses towards GA application [3]. In our findings, some genes showed tissue-specific expression such as GASA1 and 2 (highly expressed in leaves of both cultivars), whereas GASA9 and 10 showed high expression in the fruit and seed of both cultivars (Figure 6). For example, two paralogous genes, GASA6 and 7, were significantly highly expressed during all phases of ovule development in Red globe compared to Thompson seedless. This suggests that these two genes may have a role in seed development. Our observations are justified by previous findings, as TaGASR7 (Accession number AHM24216), the orthologous gene of VvGASA7, has been well-studied in wheat for its role in grain size length [20]. In *Arabidopsis*, AtGASA4 (AT5G15230), which is highly similar to VvGASA7, has its role in flower meristem development and positively regulates seed size and yield [11]. Apart from these, the orthologous gene of VvGASA7 in rice (OsGASR7; Os06g0266800) also affected grain seed length, suggesting its role in seed development [37]. Therefore, our findings suggest that some grapevine GASA genes may have a role in seed development. The functional characterization of *VvGAS7* will help scientists in exploring the mechanism of seed development in grapevine in future studies. 

In contrast to VvGAS7, during seed development stages, VvGASA5 showed high expression in Thompson seedless and almost undetectable expression in Red globe, suggesting its role in ovule abortion. High expression of VvGASA2 during different stages of seed development in seeded cultivars also supports our hypothesis that GASA genes may have a role in seed development. Li et al. [38] reported that knock-down of OsGASR9 (which is homologous of VvGASA2) reduced plant height, seed size, and overall plant yield, whereas its overexpression also increased seed size, plant height, and yield. They reported that these findings are due to the involvement of GASA genes in GA pathways. Most of the GASA genes are involved in the hormonal signaling pathway, especially in GA and ABA, and influence many plant functions, e.g., bud dormancy, seed size, and yield. Although the potential roles of genes can be predicted based on the functions of their orthologous genes in other crops, functional analyses are needed to confirm their roles in specific crops. Finally, the functional characterization of grapevine GASA genes will help scientists in understanding the molecular mechanism of seed development in grapes. 

## 4. Materials and Methods

### 4.1. Annotation and Identification of Putative Grapevine GASA Genes

We combined two complementary homology-based approaches to identify GASA genes in the grapevine genome. In the first approach, proteins annotated as GASA genes from *Arabidopsis* and apple [17] were used as queries to search open reading frame translations cataloged from a reference grapevine genome sequence (http://www.genoscope.cns.fr), Grapevine Genome CRIBI Biotech website (http://genomes.cribi.unipd.it/), and the National Centre for Biotechnology Information (NCBI; http://www.ncbi.nlm.nih.gov/), using the Basic Local Alignment Search Tool [39]. In the second method, the Hidden Markov Model (HMM) profile of the GASA domain (PFAM 02704) was used to search grapevine GASA proteins in the 12X coverage assembly of the *V. vinifera* PN40024 genome [40]. The NCBI Conserved Domain Database (https://www.ncbi.nlm.nih.gov/Structure/cdd/cdd.shtml) and Simple Modular Architecture Research Tool (SMART; http://smart.emblheidelberg.de/) were used to check the presence of the complete GASA domain in obtained protein sequences [41]. Finally, all non-redundant putative protein sequences with a conserved GASA domain were considered and used for further analysis. This approach led to the designation of 14 GASA-domain-encoding genes. 

### 4.2. Physicochemical Properties and Phylogeny Analysis

All identified VvGASA gene protein sequences, coding sequences, genomic sequences, and related information about accession number, start–end position of the gene, the number of amino acids, and chromosome location were downloaded from Grape Genome Database and NCBI. Information about the physiochemical properties of GASA proteins was obtained from the online ExPASy program (http://web.expasy.org/protparam/) using protein sequences [42]. In silico analysis of subcellular location and tertiary structure of proteins was performed using online programs The WOLF PSORT II program (http://www.genscript.com/wolfpsort.html) [43] and PHYRE server v2.0 (http://www.sbg.bio.ic.ac.uk/phyre2/html/page.cgi?id=index), respectively. Multiple sequence alignment of GASA proteins was completed using DNAMAN (Version 8, Lynnon Bio-soft, Canada) with default parameters. The phylogenetic trees including 55 proteins or with 14 GASA grapevine proteins were constructed with MEGA 5.0 software by using the following parameters: the neighbor-joining (NJ) method, ‘W’ approach for sequence alignments, 1000 bootstrap iterations, “p-distance”, “Complete Deletion”, and gap setting [44]. The phylogenetic tree among different plant species included 14, 26, and 15 protein sequences from grapes, apple, and *Arabidopsis*, respectively.

### 4.3. Exon–Intron, Gene Structures, Conserved Motif, and Promoter Analysis

Exon–intron study of the VvGASA genes was performed using aligned coding sequences and genomic sequences in the online Gene Structure Display Server 2.071 (http://gsds.cbi.pku.edu.cn/index.php). MEME 4.11.2 (http://meme-suite.org/tools/meme), an online program, was used to find up to ten conserved motifs [45]. For cis-acting elements analysis, the upstream sequence (1.5-kb) of each gene was examined through the PlantCARE (http://bioinformatics.psb.ugent.be/webtools/plantcare/html/) online program.

### 4.4. Synteny Analysis and Calculation of Ka/Ks Ratio for Duplicated Genes

The tandem and segmental duplication of genes were calculated according to their physical position on individual chromosomes. Two or more genes present on the same chromosome within a 200 kb region were considered as tandemly duplicated [46]. For segmental duplication, data were retrieved from the Plant Genome Duplication Database (http://chibba.agtec.uga.edu/duplication/) [47] and a diagram was generated using the circos program, version 0.63 (http://circos.ca/). The Ka (non-synonymous substitution rate) and Ks (synonymous substitution rate) of duplicated genes were determined using an online tool (http://services.cbu.uib.no/tools/kaks). The ratio of Ka/Ks was used to estimate the selection pressure mode [48]. The Ka/Ks ratio can reveal three different situations: positive (Ka/Ks > 1), negative (Ka/Ks < 1), and neutral (Ka/Ks = 1) [49].

### 4.5. Plant Materials

In this experiment, plant samples were collected from two grape cultivars, Red globe (Seeded cultivar) and Thompson seedless (seedless cultivar), grown under natural field conditions in the grape orchard of Northwest A&F University, Yangling, China (34°200′ N 108°240′ E). All the samples, including young leaves, tendrils, stems, flowers, and fruits (42 DAF, days after full bloom), were collected from the healthy plants. Apart from these, seed samples were taken at different developmental stages of fruits at 10, 27, 34, 40, and 50 DAF. After collection, all plant parts were immediately frozen in liquid nitrogen and preserved at −80 ℃ for RNA extraction.

### 4.6. Total RNA Extraction and Expression Analysis by RT-PCR

The EZNA Plant RNA Kit (R6827-01, OMEGA Biotek, Norcross, GA, USA) and a Nano Drop Spectrophotometer (Thermo Fisher Scientific, Yokohama, Japan) were used to extract and quantify RNA, respectively. Prime Script RTase (Trans Gen Biotech, Beijing, China) was used to synthesize first-strand cDNA from extracted RNA; cDNA was diluted six times and stored at −40 °C for future study. Primer Premier 7.0 (Appendix A) was used to design gene-specific primers. Quantitative RT-PCR was carried out for selected genes using SYBR Green (Trans Gen Biotech, Beijing, China) on an IQ5 real-time PCR machine (Bio-Rad, Hercules, CA, USA). The total reaction mixture was 20 µL consisting of 10 µL SYBR green, 7 µL sterile distilled water, 1 µL of cDNA template, 0.8 µL each primer (1.0 µM), and 0.4 µL of Rox reference dye1. The reaction was executed with the following parameters: 95 °C for 30 s, followed by 42 cycles of 95 °C for 10 s and 60 °C for 30 s. The transcript level was normalized by using the *VvActin* gene as an internal control. Each reaction was carried out with three technical and biological replicates. The comparative CT method, also known as the 2^−ΔΔCT^method, was used to calculate relative expression levels where ΔΔ*C*T = [(CT target gene – CT control gene) Sample A – (CT target gene – CT control gene) Sample B] [50]. Sigma Plot 12.5 was used to draw graphs.

## Figures and Tables

**Figure 1 ijms-21-01088-f001:**
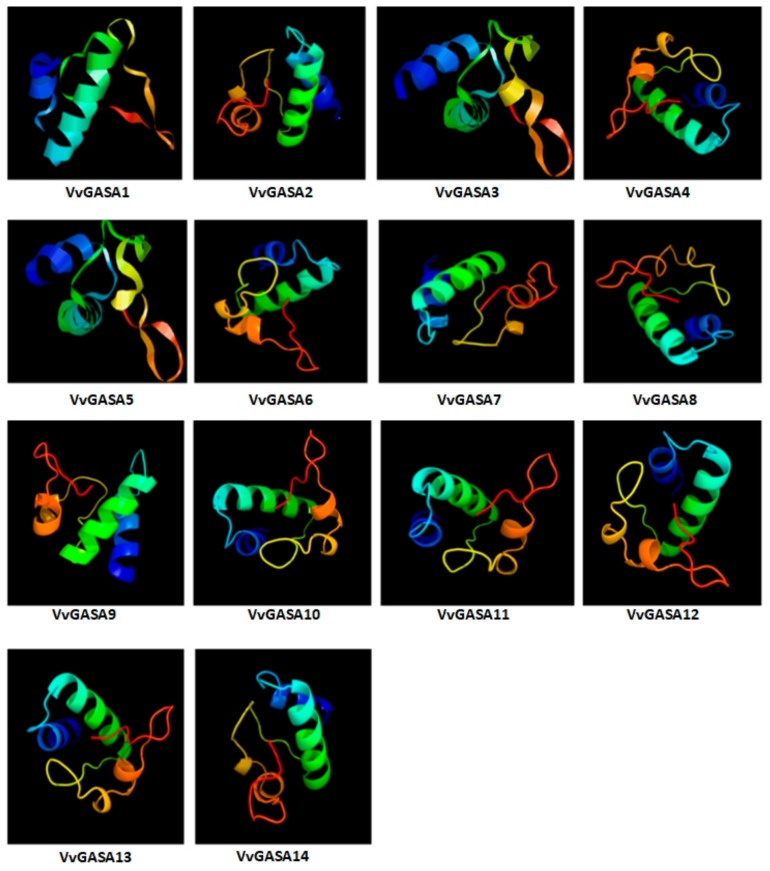
Predicted 3-D structures of GASA proteins.

**Figure 2 ijms-21-01088-f002:**
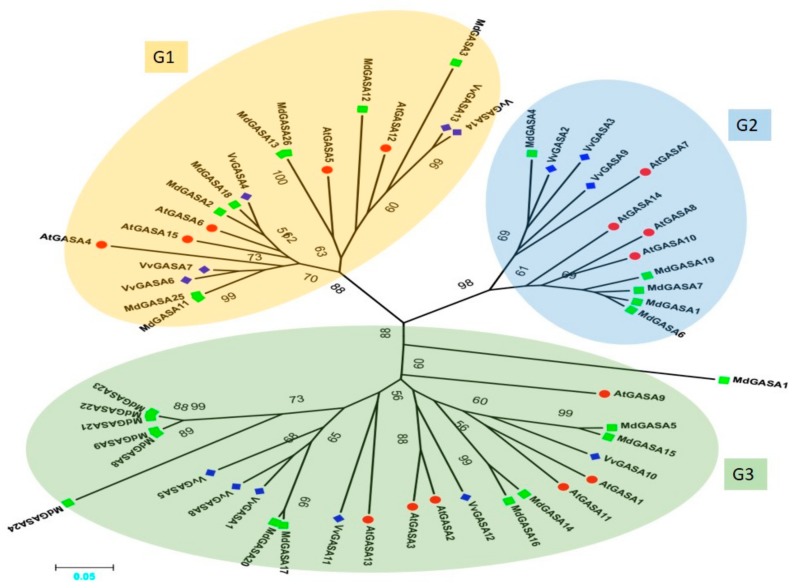
Phylogenetic tree of GASA genes of *Vitis vinifera*, *Malus domestica*, and *Arabidopsis thaliana*. Blue-colored diamonds represent grapevine protein, green-colored squares represent apple proteins, and red-colored circles represent *Arabidopsis* proteins. Different colored oval shapes indicate different groups. Numbers near the tree branches indicate bootstrap values (BS) and BS values less than 50 are not shown.

**Figure 3 ijms-21-01088-f003:**
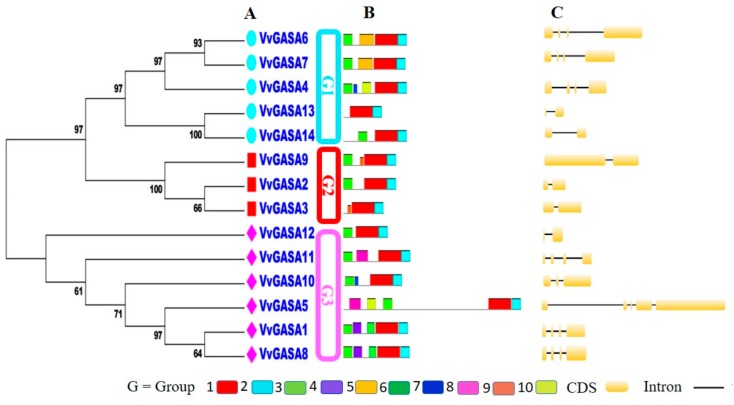
Analysis of grapevine GASA genes. (**A**) Phylogenetic tree of grapevine GASA genes. Different boxes are colored to indicating different groups. Numbers near the tree branches indicate bootstrap values. (**B**) Motif analysis. The different colors of boxes denote different motif numbers. The length of box indicates motif length. (**C**) Exon-intron distribution. CDS denotes exons.

**Figure 4 ijms-21-01088-f004:**
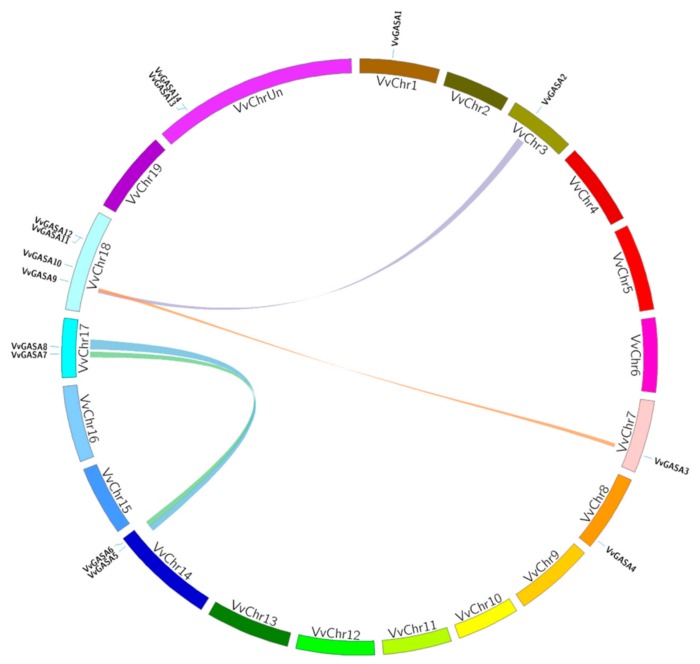
Chromosomal distribution and synteny analysis of grapevine GASA genes. Syntenic regions and chromosomal regions are depicted in different colors (Chr: chromosomes).

**Figure 5 ijms-21-01088-f005:**
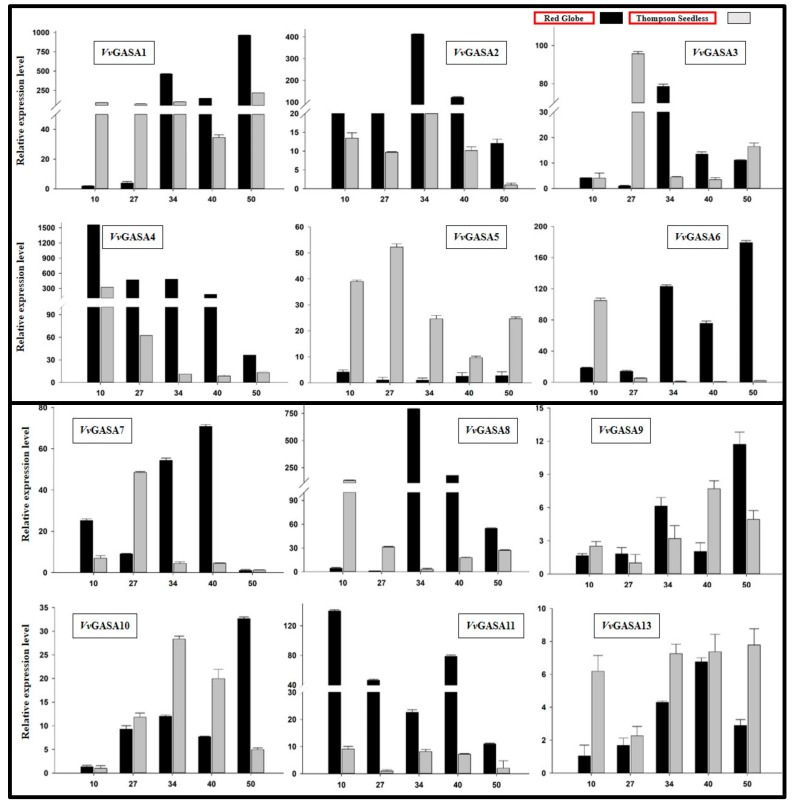
Real-time PCR analysis of grapevine GASA genes at different stages of seed development in seedless and seeded cultivar. Numbers on *x*-axis denote number of days after full bloom (DAF).

**Figure 6 ijms-21-01088-f006:**
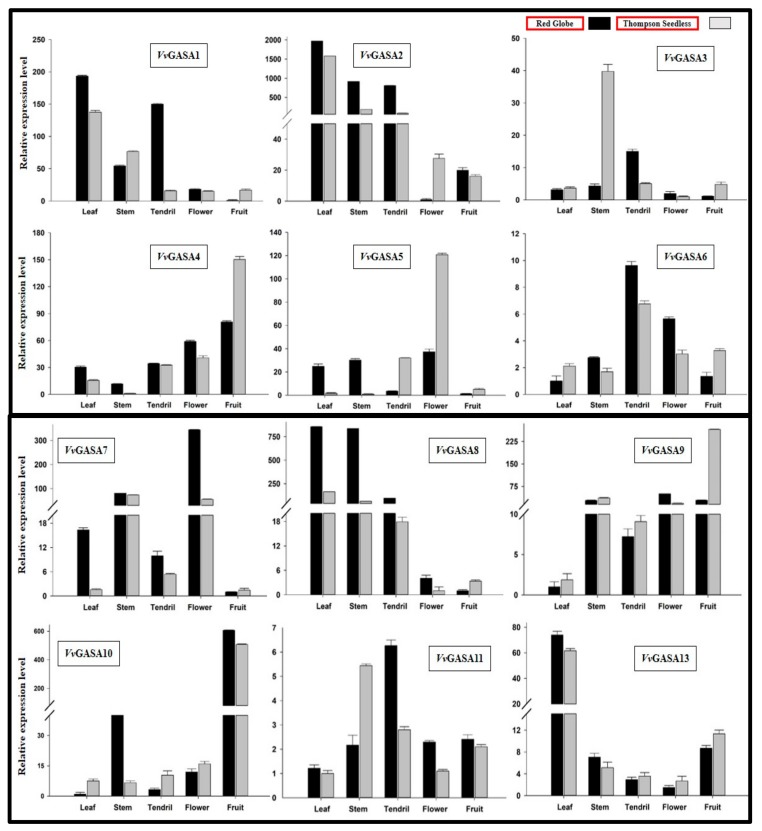
Real-time PCR analysis of different plant parts.

**Figure 7 ijms-21-01088-f007:**
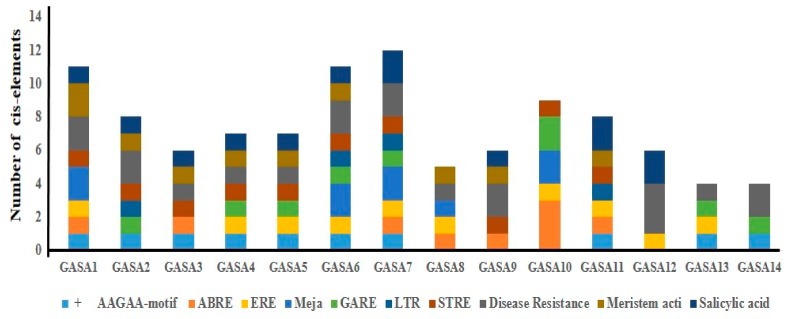
*Cis*-element prediction in the VvGASA promoters.

**Table 1 ijms-21-01088-t001:** Detailed information of grapevine GASA (gibberellic acid-stimulated Arabidopsis) genes.

Gene Locus ID	Gene ID	Accession No.	Chromosome No.	Start Site	End Site	CDS (bp)	ORF (aa)
GSVIVT01020178001	*VvGASA*1	CBI32100	1+	9381743	9382568	327	108
GSVIVT01037887001	*VvGASA*2	CBI22497	3-	6715491	6716068	267	88
GSVIVT01000168001	*VvGASA*3	CBI33733	7+	15821751	15822734	204	67
GSVIVT01033563001	*VvGASA*4	CBI30071	8-	19734994	19736181	321	106
GSVIVT01032528001	*VvGASA*5	CBI34969	14+	28107504	28112390	897	298
GSVIVT01011412001	*VvGASA*6	CBI22214	14-	29446051	29447932	321	106
GSVIVT01008003001	*VvGASA7*	CBI15224	17+	6769396	6770752	315	104
GSVIVT01007817001	*VvGASA*8	CBI15083	17+	8741561	8742409	336	111
GSVIVT01009384001	*VvGASA*9	CBI19434	18+	7913344	7915123	267	88
GSVIVT01009902001	*VvGASA*10	CBI19861	18-	12275798	12276616	297	98
GSVIVT01034477001	*VvGASA*11	CBI18167	18+	20718815	20720279	339	112
GSVIVT01034476001	*VvGASA*12	CBI18166	18-	20720304	20720676	225	74
GSVIVT01003387001	*VvGASA*13	CBI25689	Un-	9775242	9775609	195	64
GSVIVT01003388001	*VvGASA*14	CBI25690	Un-	9791751	9792551	321	106

CDS: coding sequence; Chr: chromosome; ORF: open reading frame; Un: unknown chromosome.

**Table 2 ijms-21-01088-t002:** Amino acid composition and physiochemical characteristics of GASA proteins.

Gene	MW	PI	Major Amino Acid%	Instability Index	Aliphatic Index	GRAVY	Localization Predicted
GASA1	11.96	8.61	C(11.9), L(8.3), R(7.3)	36.64	67.06	−0.172	extr., vacu.
GASA2	9.71	9.02	C(14.8), K(12.5), L(11.4)	38.49	58.75	−0.281	chlo, nucl., extr
GASA3	7.28	8.87	C(17.9), K(13.4), G(10.4)	41.98	33.43	−0.515	chlo., nucl., cyto., extr.
GASA4	11.85	9.20	C(11.2), P(11.2), T(9.3)	51.40	45.61	−0.421	extr., chlo., nucl.
GASA5	31.96	9.64	P(23.7), V(9.7), K(8.4)	67.94	78.53	−0.241	cyto., ER
GASA6	11.79	9.30	C(11.3), K(11.3), G(9.4)	35.75	53.40	−0.289	extr., vacu., chlo.
GASA7	11.62	9.22	C(11.4), K(11.4), P (8.6)	38.76	57.62	−0.233	extr., vacu.
GASA8	12.27	8.66	C(10.7), L(9.8), G(8.9)	45.33	80.80	0.046	extr., chlo., vacu.
GASA9	9.71	8.96	C(13.5), K(13.5), S(10.1)	44.76	54.83	−0.206	extr., chlo., vacu.
GASA10	10.35	8.50	C(12.1), A(10.1), S(10.1)	47.05	63.23	−0.143	extr., vacu.
GASA11	12.62	9.52	S(11.5), C(10.6), K(10.6)	50.04	68.23	−0.344	extr.
GASA12	8.36	9.00	C(16), K(10.7), A(10.7)	42.75	49.47	−0.417	mito., chlo., cyto.
GASA13	7.42	8.50	C(17.2), K(10.9), Y(9.4)	42.31	41.09	−0.492	nucl., cyto., mito.
GASA14	11.96	8.80	C(12.3), L(12.3), K(11.3)	45.71	81.79	0.103	chlo., nucl., extr.

MW: molecular weight (kDa), pI: isoelectric point, GRAVY: grand average of hydropathicity,A: Ala, R: Arg, C: Cys, G: Gly, L: Leu, K: Lys, P: Pro, S: Ser, T: Thr, Y: Tyr, Extra: extracellular, Vacu: vacuoles, Chlo: chloroplast, Cyto: cytoplasm, Mito: mitochondria, Nucl: nucleus, Plas: plastids, and ER endoplasmic reticulum.

**Table 3 ijms-21-01088-t003:** Duplications of GASA genes in grapes.

Gene1	Gene2	Ka	Ks	Ka/Ks	Selection Pressure	Gene Duplications
*GASA7*	*GASA6*	0.156	1.5871	0.0928	Purifying selection	Segmental
*GASA8*	*GASA5*	0.4568	1	0.4568	Purifying selection	Segmental
*GASA9*	*GASA2*	0.1985	0.929	0.213	Purifying selection	Segmental
*GASA3*	*GASA9*	0.1781	1.1731	0.151	Purifying selection	Segmental
*GASA12*	*GASA11*	0.196	0.333	0.585	Purifying selection	Tandem
*GASA14*	*GASA13*	0.040	0.074	0.540	Purifying selection	Tandem

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
