# Peer review of "Genome-Wide Characterization and Expression Profiling of GASA Genes during Different Stages of Seed Development in Grapevine (Vitis vinifera L.) Predict Their Involvement in Seed Development"

_ijms, 2020, doi:10.3390/ijms21031088_

Round 1

Reviewer 1 Report

The manuscript “Genome-wide characterization and expression 2 profiling of GASA genes during different stages of 3 seed development in Grapevine (Vitis vinifera L.) 4 predict their involvement in seed development” describes the GASA genes in grapevine, their structure, phylogenetic relationships and expression profiling across tissues with emphasis on fruit development. The authors also compare GASA gene expression between seeded and seedless grapevine cultivars. The manuscript is highly descriptive, but it has a potential to bring useful information for both grapevine breeders and plant biologists in general.

Main issues

1.The authors observed contrasting expression of VvGASA7 and VvGASA5 in seedless versus seeded grapevine cultivars. It would be valuable to supplement more information about those genes, for example on their response to exogenous GA application, which was studied in e.g. Arabidopsis thaliana. The GA content measurements in the tissues where GASA expression was estimated would be very useful. It is difficult to make conclusion about possible gene function just on the basis of gene expression in various tissues, developmental stages and cultivars, without additional experiments.

There are numerous points in the text, which are difficult to understand.

2. l. 121-122: “According to results, most 121 of the grapevine GASA genes were found in extra-extracellular organelles of the cell with a little share 122 in vacuole, chloroplast, and cytoplasm (Table 2).”

  Extracellular or organelles?

3. l.136-136. It is not necessary to explicitly discuss the number of genes in the three group. In contrast, the description of distinction (individual motifs) shall be made clearer.

4. l. 158-159. I do not understand the statement. “According to the results some similar proteins have similar motifs arrangements with respect to either motif number or gene length (e.g. VvGASA6/7, and VvGASA1/8).”

Is not it redundant? Please, avoid a repetitive usage of the phrase “According to the results” throughout the text.

5. l. 257-260. “This shows that grapevine and Arabidopsis were closely related before 257 speciation and the number of genes in GASA family is highly conserved.”

This sentence does not make sense !

6. Phylogeny analysis l. 342-347. NJ method is fast, but less accurate. Please, use also MP or Ml approach.

7. RT qPCR l. 385. Please, estimate PCR efficiency and use it in the calculation of relative expression.

Minor point

The writing quality is not sufficiently good, there are typos and errors in style and grammar, the text is sometimes repetitive. Please, ask a native speaker to correct the text!

Author Response

Dear Reviewer,

Thank you for moderating the review of our manuscript, ‘Genome-wide characterization and expression profiling of GASA genes during different stages of seed development in Grapevine (Vitis vinifera L.) predict their involvement in seed development (Manuscript ID: IJMS 695314)’. We appreciate the opportunity to submit a revised manuscript that we feel adequately addresses your comments. A point-by-point response to comments is below. We hope that with these changes, the manuscript is now suitable for publication in International Journal of Molecular Sciences. We would be happy to consider any further changes that you deem appropriate.

Thank you again, and best regards,

Xiping Wang

Prof and Dr.

College of Horticulture

Northwest A&F University

Main issues

The authors observed contrasting expression of VvGASA7 and VvGASA5 in seedless versus seeded grapevine cultivars. It would be valuable to supplement more information about those genes, for example on their response to exogenous GA application, which was studied in e.g. Arabidopsis thaliana. The GA content measurements in the tissues where GASA expression was estimated would be very useful. It is difficult to make conclusion about possible gene function just on the basis of gene expression in various tissues, developmental stages and cultivars, without additional experiments.

Response: Thanks for the suggestion. In our studies, apart from performing expression analysis, we have also performed different types of bioinformatic analysis for example cis-element analysis and subcellular localization. We surmise our study provides basic genomic information and inklings about the functions of the grapevine GASA gene family. Furthermore, it will be useful in selecting candidate genes related to seed development in grapevine and pave the way for further functional verification of these GASA genes. So, keeping in view the postulated hypothesis and designated objectives to be achieved, it is perhaps a different aspect which cannot be performed at the moment owing to technical glitches. However, suggestion has been seriously noted to be incorporated in upcoming study design and answer will be executed as per line in next works.

There are numerous points in the text, which are difficult to understand.

l. 121-122: “According to results, most 121 of the grapevine GASA genes were found in extra-extracellular organelles of the cell with a little share 122 in vacuole, chloroplast, and cytoplasm (Table 2).”

Extracellular or organelles?

Response: Thank you for the advice, this is extracellular (L131).

l.136-136. It is not necessary to explicitly discuss the number of genes in the three group. In contrast, the description of distinction (individual motifs) shall be made clearer.

 Response: Thank you for the advice, changed according to suggestion. We have added new lines (147-148 &174-179) in the revised M.S.

l. 158-159. I do not understand the statement. “According to the results some similar proteins have similar motifs arrangements with respect to either motif number or gene length (e.g. VvGASA6/7, and VvGASA1/8).”

Response: Thank you for pointing this out, corrections have been made in the revised version.  Paralogous gene pairs (VvGASA6/VvGASA7 and VvGASA1/VvGASA8) have almost similar motif arrangements (L178).

Is not it redundant? Please, avoid a repetitive usage of the phrase “According to the results” throughout the text.

 Response: Thank you for the advice, changed according to the suggestion.

l. 257-260. “This shows that grapevine and Arabidopsis were closely related before 257 speciation and the number of genes in GASA family is highly conserved.”

This sentence does not make sense!

Response: Thanks for the suggestion. We have reframed the sentence like this, perhaps grape is more similar to Arabidopsis than to apple (L328-329).

Phylogeny analysis l. 342-347. NJ method is fast, but less accurate. Please, use also MP or Ml approach.

Response: Thank you for the advice. According to suggestion, we have made new Phylogenetic trees with Maximum likelihood method (ML) and MP. Moreover, the Phylogenetic tree generated using the ML method with 1000 bootstrap replicates based on the JTT matrix-based model is given below for your kind consideration.

For image kindly see the attached file.

RT qPCR l. 385. Please, estimate PCR efficiency and use it in the calculation of relative expression.

Response: Thank you for pointing this out. The quantitative endpoint for real-time PCR is the threshold cycle (CT). The CT is defined as the PCR cycle at which the fluorescent signal of the reporter dye crosses an arbitrarily placed threshold. The numerical value of the CT is inversely related to the amount of amplicon in the reaction (i.e., the lower the CT, the greater the amount of amplicon). Relative expression levels were determined by the comparative CT method also referred to as 2 −ΔΔCT method, where ΔΔCT= [(CT target gene – CT control gene) Sample A – (CT target gene – CT control gene) Sample B]. Whereas, Sample A, denotes target sample and Sample B is control sample (Thomas & Kenneth, 2008, Nature Protocols). It has been added in material and methods (L461-462).

Minor point

The writing quality is not sufficiently good, there are typos and errors in style and grammar, the text is sometimes repetitive. Please, ask a native speaker to correct the text!

Response: Thank you for this advice. We have corrected all the grammatical and contextual mistakes.

Reviewer 2 Report

Overview:

In the manuscript “Genome-wide characterization and expression profiling of GASA genes during different stages of seed development in Grapevine (Vitis vinifera L.) predict their involvement in seed development” the authors surveyed the Vitis genome and found 14 putative GASA genes. They used several approaches including phylogenetics to characterize the putative genes. Additionally, the authors functionally characterized GASA genes in two cultivars of Vitis throughout several tissues and life stages. The authors did an extensive bioinformatic characterization of the gene family in addition to looking at expression in different tissues using qRT-PCR. Overall the paper is well organized and easy to follow. While the study design and analyses are common, the characterization of GASA in Vitis is an important piece of understanding the role of GASA genes in Angiosperms.

Major comments:

Lines 28 and 29. I would say the proteins were predicted to be in the extracellular region since the analysis was bioinformatic in nature and not in vivo.

Section 2.1. Genome-wide identification and protein features of GASA genes in grapevine

A summary sentence including information on how many putative genes were identified in the original queries and how many were filtered out at each step is necessary.

Lines 99-101. These are methods.

Lines 132 and 133 – this information on phylogenetic sampling needs to be in the methods as well and accession information for that sequence data needs to be included.

Lines 136 and 137 – what do lowest and highest refer to? These are not phylogenetic terms. Are G1,G2, and G3 determined because they for monophyletic clades??? Because there is no phylogenetic support for G3 being a separate clade from G2 in this phylogeny. Please look at González et al. 2015 for an example of how to talk about gene families in a phylogenetic context. DOI: https://doi.org/10.1007/s11295-015-0871-0

Line 142 – the larger phylogeny with outgroups does not support a separate clade for G3. The Vitis specific phylogeny does though.

Lines 145 and 146 – These data show that the gene is more structurally conserved. Conserved in this context can be confusing for the reader, not meaning phylogenetic conservation necessarily or sequence similarity but a structural similarity. Clarifying the meaning of conserved in this context will help the reader understand the argument more.

Section 2.3 – This section would be strengthened by adding a similarity value to individuals within each clade. Not only are the introns in the same location retained but the proteins are have a sequence similarity of??? This would give additional support to the claims that certain clades are more conserved.

Lines 155–157 Figure 2 legend needs to contain information on what the colors of the shapes are and what the numerical values stand for. Additionally, collapsing all nodes at a BS value less than 50 is standard practice and will actually make the pattern

Lines 171 and 172 (Fig. 3 legend) – State that the color indicates which motif is present and the schematic shows the order. Also, there is a terminology switch from exon to CDS – would be clearer to use exon in the legend since that is the term most frequently used in the text.

Lines 254-256 – Revise this - the phylogeny only supports three groups. The Vitis phylogeny supports three groups, not the broader phylogeny with outgroups.

Lines 256 and 257 – I do not think you can make a broad statement about the close relationship between grape and Arabidopsis based on the gene copy number alone. The loss of one copy may be common and we do not have the data to know. This statement needs to be reframed. Perhaps grape is more similar to Arabidopsis than to apple etc.

Line 260 – I am still confused about what the authors mean by highest and lowest genes. Molecular weight?

Section 4.1. Annotation and identification of Putative grapevine GASA genes: Simply having the presence of a domain does not mean the gene sequence is no longer putative. The statement of manually checking the sequences identified with a domain finder needs to be clarified by changing the phrasing to indicate that these are still putative and are further tested via phylogenetics and expression analyses followed to validate the gene status.

Need to add clarification as to what went into the phylogenetic analysis. What comparable sequences were included and from what species (e.g. Arabidopsis, etc.). Does the total of 65 accessions include the 14 putative grape GASA genes or is this in addition? There needs to be accession information and references given for the sequences not generated by this study included in a table in the paper.

Was the alignment made from nucleotide or amino acid sequences? Was there any adjustment of the alignment by eye to ensure that protein motif homology was accurate? Additionally, there are extremely low bootstrap values reported on the phylogeny in Fig. 2. These need to be collapsed because a node with a value of 6 does not represent a real bifurcation. Just because the NJ method reports a bifurcation does not mean it is “real”, which is why bootstrap support is assigned. The easiest way to collapse a node based on BS value is to use TreeGraph2 (http://treegraph.bioinfweb.info).

Minor comments:

The first sentence of the second paragraph would be a better introductory sentence. As the current first sentence of the manuscript is a run on of abbreviations and parentheticals without context. Then change the order of information slightly so that the broader information about GASA comes first followed by the more specific information about naming and the number of amino acids in each characterized region.

Lines 84 and 85. “In recent year, scientists have reported some genes having a role in seed development. However, the key genes mediating this process still needed to be explored.” Which genes? What species? A “For example, …” type of statement here would help fill in the introduction of the GASA gene family as one of the key gene families involved.

Lines 133 and 134 – “The analysis included 65 GASA genes comprising of 14, 10, 26 and 15 from grapes, rice, 134 apple, and Arabidopsis, respectively.” This is methods and needs to be stated there.

Line 149 – “lengths were closely related siblings” is a confusing term. Do the authors mean paralogs? Orthologs? The terminology of phylogenetic relationships needs to be more precise.

Line 175 – Change “Chromosomes 18 have the highest, four genes (35%).” to something like “Chromosome 18 has a larger proportion of GASA genes (4; 35%) than any other chromosome.”

Line 183 – “and of” should just be of

Line 187 – replacer the comma with “a” so that the sentence reads “…belonged to the same group suggesting a common ancestor.”

In Figs. 5 & 6 the legend shows Thompsons seedless as white bars but they are gray in the figures.

Lines 228 and 229 – Changing this phrasing “However, cis-elements related to Ethylene, Gibberellic acid and Salicylic acid were more in number.” to “However, there were more cis-elements related to Ethylene, Gibberellic acid and Salicylic acid.” would improve the readability.

Lines 274 and 275 – It is not critical to address this but I am curious if the authors think this is subfunctionalization or not.

Line 332. No redundant should be changed to non-redundant.

Section 4.2. Physicochemical Properties and Phylogeny Analysis Line 335 – The aside (protein, coding and genomic sequences) doesn’t make sense in the context of identifying gene sequences. Do the author’s mean to say that all the sequence data from their queries were downloaded? If so maybe just delete the parentheticals?

Line 357. Sentences should not start with Arabic numbers that aren’t spelled. Two instead of 2.

Line 358. A comma after duplication would increase readability.

Author Response

Dear Reviewer,

Thank you for moderating the review of our manuscript, ‘Genome-wide characterization and expression profiling of GASA genes during different stages of seed development in Grapevine (Vitis vinifera L.) predict their involvement in seed development (Manuscript ID: IJMS 695314)’. We appreciate the opportunity to submit a revised manuscript that we feel adequately addresses your comments. A point-by-point response to these comments is below. We hope that with these changes, the manuscript is now suitable for publication in International Journal of Molecular Sciences. We would be happy to consider any further changes that you deem appropriate.

Thank you again, and best regards,

Xiping Wang

Prof and Dr.

College of Horticulture

Northwest A&F University

Major comments:

Lines 28 and 29. I would say the proteins were predicted to be in the extracellular region since the analysis was bioinformatic in nature and not in vivo.

Response: Thank you for the advice, revised according to suggestion.

Section 2.1. Genome-wide identification and protein features of GASA genes in grapevine

A summary sentence including information on how many putative genes were identified in the original queries and how many were filtered out at each step is necessary.

Response: Thank you for pointing this out, revised according to suggestion (L407 &408).

Lines 99-101. These are methods.

Response: Thanks for suggestion, changed.

Lines 132 and 133 – this information on phylogenetic sampling needs to be in the methods as well and accession information for that sequence data needs to be included.

Response: Thank you for the advice, revised according to suggestion and accession information has been added in the supplementary data of revised version (TableS5).

Lines 136 and 137 – what do lowest and highest refer to? These are not phylogenetic terms. Are G1, G2, and G3 determined because they for monophyletic clades??? Because there is no phylogenetic support for G3 being a separate clade from G2 in this phylogeny. Please look at González et al. 2015 for an example of how to talk about gene families in a phylogenetic context. DOI: https://doi.org/10.1007/s11295-015-0871-0

Response: Thank you for pointing this out. Actually according to previous studies in different crops, researchers have divided GASA gene family in different groups. For example Arabidopsis, apple and soybean GASA gene family has been divided into three groups and named as G1, G2, and G3 [Fan et al.2017; Ahmad et al. 2019]. Whereas, Bang et al. (2018) divided GASA gene family members (rice, rubber plant and Arabidopsis) into two classes. Furthermore, Muhammad et al. (2019) classified ten plants species (monocots and dicots) into well conserved four subgroups. As far as, we know mostly people have followed the classification of Fan et al, so to fix the classification issue, we have made a new phylogenetic tree which includes grapes, Arabidops, and apple. Moreover tree is generated using N.J method keeping bootstrap value 1000 and is presented in the branch form.

For figure kindly see the attached file.

Line 142 – the larger phylogeny with out-groups does not support a separate clade for G3. The Vitis specific phylogeny does though.

Response: Thanks for the suggestion. We have made a new phylogenetic to fix the issue.

Lines 145 and 146 – These data show that the gene is more structurally conserved. Conserved in this context can be confusing for the reader, not meaning phylogenetic conservation necessarily or sequence similarity but a structural similarity. Clarifying the meaning of conserved in this context will help the reader understand the argument more.

Response: Thanks for this advice, we have revised the sentence (L155).

Section 2.3 – This section would be strengthened by adding a similarity value to individuals within each clade. Not only are the introns in the same location retained but the proteins are have a sequence similarity of??? This would give additional support to the claims that certain clades are more conserved.

Response: Thanks for the suggestion. We have added new lines according to suggestion for example L147-148, L159-163, &174-179.

Lines 155–157 Figure 2 legend needs to contain information on what the colors of the shapes are and what the numerical values stand for. Additionally, collapsing all nodes at a BS value less than 50 is standard practice and will actually make the pattern

Response: Thank you for pointing this out. Changings has been made according to suggestion (L171-173).

Lines 171 and 172 (Fig. 3 legend) – State that the color indicates which motif is present and the schematic shows the order. Also, there is a terminology switch from exon to CDS – would be clearer to use exon in the legend since that is the term most frequently used in the text.

Response: Thank you for pointing this out. Changed accordingly in the revised manuscript (L192-194).

Lines 254-256 – Revise this - the phylogeny only supports three groups. The Vitis phylogeny supports three groups, not the broader phylogeny with outgroups.

Response: Thank you for pointing this out. We have revised it in the revised manuscript.

Lines 256 and 257 – I do not think you can make a broad statement about the close relationship between grape and Arabidopsis based on the gene copy number alone. The loss of one copy may be common and we do not have the data to know. This statement needs to be reframed. Perhaps grape is more similar to Arabidopsis than to apple etc.

Response: Thank you for pointing this out, correction has been made (L328-329).

Line 260 – I am still confused about what the authors mean by highest and lowest genes. Molecular weight?

Response: Thanks for the suggestion. We have revised the sentence to clarify our view point.

Section 4.1. Annotation and identification of Putative grapevine GASA genes: Simply having the presence of a domain does not mean the gene sequence is no longer putative. The statement of manually checking the sequences identified with a domain finder needs to be clarified by changing the phrasing to indicate that these are still putative and are further tested via phylogenetics and expression analyses followed to validate the gene status.

Response: Thank you for pointing this out, corrections have been made in the revised version (L406).

Need to add clarification as to what went into the phylogenetic analysis. What comparable sequences were included and from what species (e.g. Arabidopsis, etc.). Does the total of 65 accessions include the 14 putative grape GASA genes or is this in addition? There needs to be accession information and references given for the sequences not generated by this study included in a table in the paper.

Response: Thanks for the suggestion. We have provided complete accession information about species in the supplementary data of revised version (Table S4 & S5).

Was the alignment made from nucleotide or amino acid sequences? Was there any adjustment of the alignment by eye to ensure that protein motif homology was accurate? Additionally, there are extremely low bootstrap values reported on the phylogeny in Fig. 2. These need to be collapsed because a node with a value of 6 does not represent a real bifurcation. Just because the NJ method reports a bifurcation does not mean it is “real”, which is why bootstrap support is assigned. The easiest way to collapse a node based on BS value is to use TreeGraph2 (http://treegraph.bioinfweb.info).

Response: Thanks for the suggestion. The alignment was made from amino acid sequences and no adjustment has been made. We have already provided figure of multiple sequence alignment in the supplementary date (S1). Moreover, we have collapsed B.S values less than 50 in revised version.

Minor comments:

The first sentence of the second paragraph would be a better introductory sentence. As the current first sentence of the manuscript is a run on of abbreviations and parentheticals without context. Then change the order of information slightly so that the broader information about GASA comes first followed by the more specific information about naming and the number of amino acids in each characterized region.

Response: Thank you for pointing this out, changes has been made in the revised version

Lines 84 and 85. “In recent year, scientists have reported some genes having a role in seed development. However, the key genes mediating this process still needed to be explored.” Which genes? What species? A “For example, …” type of statement here would help fill in the introduction of the GASA gene family as one of the key gene families involved.

Response: Thank you for the advice, revised according to suggestion (L93-94).

Lines 133 and 134 – “The analysis included 65 GASA genes comprising of 14, 10, 26 and 15 from grapes, rice, 134 apple, and Arabidopsis, respectively.” This is methods and needs to be stated there.

Response: Thank you for the advice, changed according to suggestion (L421-422).

Line 149 – “lengths were closely related siblings” is a confusing term. Do the authors mean paralogs? Orthologs? The terminology of phylogenetic relationships needs to be more precise.

Response: Thank you for pointing this out, we have replaced siblings with paralogs gene pairs.

Line 175 – Change “Chromosomes 18 have the highest, four genes (35%).” to something like “Chromosome 18 has a larger proportion of GASA genes (4; 35%) than any other chromosome.”

Response: Thanks for the suggestion we have revised the sentence according to suggestion (L 197).

Line 183 – “and of” should just be of

Response: Thank you for pointing this out, done as suggested.

Line 187 – replacer the comma with “a” so that the sentence reads “…belonged to the same group suggesting a common ancestor.”

Response: Thank you for pointing this out, done as suggested.

In Figs. 5 & 6 the legend shows Thompsons seedless as white bars but they are gray in the figures.

Response: Thank you for pointing this out, corrections have been made in the revised version (L223 &L257).

Lines 228 and 229 – Changing this phrasing “However, cis-elements related to Ethylene, Gibberellic acid and Salicylic acid were more in number.” to “However, there were more cis-elements related to Ethylene, Gibberellic acid and Salicylic acid.” would improve the readability.

Response: Thank you for pointing this out, done as suggested (L299).

Lines 274 and 275 – It is not critical to address this but I am curious if the authors think this is subfunctionalization or not.

Response: Thank you for pointing this out, we think no.

Line 332. No redundant should be changed to non-redundant (L 406).

Response: Thank you for pointing this out, done as suggested.

Section 4.2. Physicochemical Properties and Phylogeny Analysis Line 335 – The aside (protein, coding and genomic sequences) doesn’t make sense in the context of identifying gene sequences. Do the author’s mean to say that all the sequence data from their queries were downloaded? If so maybe just delete the parentheticals?

Response: Thank you for pointing this out, done as suggested (L410).

Line 357. Sentences should not start with Arabic numbers that aren’t spelled. Two instead of 2.

Response: Thanks for the suggestion, we have revised the sentence according to suggestion (L433).

Line 358. A comma after duplication would increase readability.

Response: Thank you for the advice, changed according to suggestion (L434).

Round 2

Reviewer 1 Report

The revised manuscript “Genome-wide characterization and expression 2 profiling of GASA genes during different stages of 3 seed development in Grapevine (Vitis vinifera L.) 4 predict their involvement in seed development” was improved only a little, some points persist.

The contribution of this study without experimental estimation of GAs contents is only incremental. The authors did not consider PCR efficiency in their calculations of relative expression values. In reality, efficiency is lower than 100% and the number different from 2 (1.98) shall be used in calculations. Even small deviation from 100% may change the results, as the equation is exponential! The authors corrected the text, where it was recommended by the reviewers. However, the rest of text still needs extensive revision by a native speaker! It is often too lengthy and repetitive. Some corrections are not good. For example (l. 328-9):

“This suggests that grape is more similar to Arabidopsis than to apple “

Apple and Arabidopsis are both very distant from grapevine, this sentence brings no information.

Author Response

Dear Reviewer,

Thank you for moderating the review of our manuscript, ‘Genome-wide characterization and expression profiling of GASA genes during different stages of seed development in Grapevine (Vitis vinifera L.) predict their involvement in seed development (Manuscript ID: IJMS 695314)’. We appreciate the opportunity to submit a revised manuscript that we feel adequately addresses your comments. A point-by-point response to these comments is below. We hope that with these changes, the manuscript is now suitable for publication in International Journal of Molecular Sciences. We would be happy to consider any further changes that you deem appropriate.

Thank you again, and best regards,

Xiping Wang

Prof and Dr.

College of Horticulture

Northwest A&F University

The authors corrected the text, where it was recommended by the reviewers. However, the rest of text still needs extensive revision by a native speaker! It is often too lengthy and repetitive. Some corrections are not good. For example (l. 328-9):

“This suggests that grape is more similar to Arabidopsis than to apple “

Apple and Arabidopsis are both very distant from grapevine, this sentence brings no information.

Response: Thanks for the suggestion. We have revised the M.S according to your guidelines and have corrected all the grammatical and contextual mistakes. Prof. Dr. Habib-ur-Rehman (http://www.bzu.edu.pk/facultyindex.php?id=20) has helped us in English editing. We have acknowledged him in the acknowledgment section of M.S.

Reviewer 2 Report

The authors did a great job of revising the phylogenetic methods, including the removal of rice which was causing conflict in the dataset.

A moment to clarify why going along with what has been published previously can sometimes be misleading. I would agree that while other cited papers "found" three groups, the presentation of the referenced material does not always support this. For example, using the nomenclature put forth in Fan et al. 2017 is fine as long as we keep in mind that little statistical support is provided with the Fig. 5 phylogeny, other than four nodes with BS values of 100, and no support information is provided at all on the phylogeny in Fig. 4. This makes the data very suspect considering all other nodes could have values of 5 or 95. The other referenced papers do a much better job of presenting their phylogenies which do support three groups when monocots are excluded. Likely the reason differing numbers of groups are being found is due to taxon sampling and the NJ method itself. Long-branch attraction is probably an issue when a gene family is being characterized in monocots and eudicots, thus each dataset with varying taxon sampling supports different hypotheses of relationships. Using a a tree searching method that has optimality criterion rather than a distance-based, clustering method would likely help with some of the confusion and resolve some relationships. This understanding of underlying phylogenetic theory might help in future analyses when trying to determine how many groups of GASA or other genes there really are.

A minor suggested change for further clarity: the use of the terms maximum (previously highest) and minimum (previously lowest) to describe gene content might be more clear if the authors used the words "most" and "least" or "fewest" to describe the number of genes. This is because maximum and minimum have a connotation of being limits of the number of something while most and least/fewest are only about the counts relative the rest of the topic.

Author Response

Dear Reviewer,

Thank you for moderating the review of our manuscript, ‘Genome-wide characterization and expression profiling of GASA genes during different stages of seed development in Grapevine (Vitis vinifera L.) predict their involvement in seed development (Manuscript ID: IJMS 695314)’. We appreciate the opportunity to submit a revised manuscript that we feel adequately addresses your comments. A point-by-point response to these comments is below. We hope that with these changes, the manuscript is now suitable for publication in International Journal of Molecular Sciences. We would be happy to consider any further changes that you deem appropriate.

Thank you again, and best regards,

Xiping Wang

Prof and Dr.

College of Horticulture

Northwest A&F University

A minor suggested change for further clarity: the use of the terms maximum (previously highest) and minimum (previously lowest) to describe gene content might be more clear if the authors used the words "most" and "least" or "fewest" to describe the number of genes. This is because maximum and minimum have a connotation of being limits of the number of something while most and least/fewest are only about the counts relative the rest of the topic.

Response: Thank you for the advice, revised according to suggestion (L308 & 325).